# The free energy landscape of retroviral integration

Willem Vanderlinden [1,2]*, Tine Brouns[2,3], Philipp U. Walker[1], Pauline J. Kolbeck[1], Lukas F. Milles [1], Wolfgang Ott[1], Philipp C. Nickels[1], Zeger Debyser [3] & Jan Lipfert [1]*

Retroviral integration, the process of covalently inserting viral DNA into the host genome, is a point of no return in the replication cycle. Yet, strand transfer is intrinsically iso-energetic and it is not clear how efficient integration can be achieved. Here we investigate the dynamics of strand transfer and demonstrate that consecutive nucleoprotein intermediates interacting with a supercoiled target are increasingly stable, resulting in a net forward rate. Multivalent target interactions at discrete auxiliary interfaces render target capture irreversible, while allowing dynamic site selection. Active site binding is transient but rapidly results in strand transfer, which in turn rearranges and stabilizes the intasome in an allosteric manner. We find the resulting strand transfer complex to be mechanically stable and extremely long-lived, suggesting that a resolving agent is required in vivo.

[1] Department of Physics, Nanosystems Initiative Munich, Center for NanoScience, LMU Munich, Amalienstrasse 54, 80799 Munich, Germany. [2] KU Leuven, Division of Molecular Imaging and Photonics, Celestijnenlaan 200F, 3001 Leuven, Belgium. [3] KU Leuven, Department of Pharmaceutical and Pharmacological Sciences, Kapucijnenvoer 33 blok i - box 7001, 3000 Leuven, Flanders, Belgium. *email: willem.vanderlinden@kuleuven.be; jan.lipfert@lmu.de

Successful retroviral infection requires integration of the viral genome into genomic host DNA. The machine responsible for integration is termed the intasome, a multimeric nucleoprotein complex of the viral enzyme integrase assembled on the viral DNA ends[1–3]. To establish integration, intasomes first catalyze the removal of both 3′-dinucleotides of the viral DNA[1,4]. The resulting cleaved intermediate (CI) associates with host DNA to form the target capture complex (TCC). In the TCC, integrase catalyzes strand transfer, i.e. transesterification of the processed viral DNA ends with the target DNA backbone[4], resulting in the strand transfer complex (STC). Finally, the STC must disassemble and endogenous cellular enzymes repair the gapped transfer product to establish the functional provirus[5–7].

In vitro reconstitution of well-defined intasomes paved the way for atomic resolution structures. Spuma-, beta-/gamma-, and lentiviral intasomes are composed of integrase tetramers, octamers, or hexadecamers, respectively[8–13]. Despite this architectural diversity, a genus-independent intasome core is conserved throughout the available structures. From this perspective, tetrameric prototype foamy virus (PFV) intasomes constitute a minimal model system. Structures of key PFV integration intermediates have been determined by X-ray crystallography[11,14,15], suggesting a mechanistic model for integration (Fig. 1a).

However, the structural evidence leaves many questions unresolved: what are the dynamics of the steps along the integration pathway and how can overall integration be efficient, given that strand transfer is intrinsically iso-energetic? A recent single-molecule study found a highly inefficient target capture and/or strand transfer, resulting in a short-lived STC[16]. These results challenge the hypothesis that efficient integration is achieved through an increasing stability of consecutive nucleoprotein intermediates[2,17–19]. It therefore remains unclear whether and how retroviruses have evolved a mechanism for a high strand-transfer yield and what processes follow strand-transfer completion.

Here we use biochemical assays, atomic force microscopy (AFM), and multiplexed single-molecule magnetic tweezers (MT) to study PFV strand-transfer dynamics. Our data directly demonstrate that consecutive intermediates form virtually irreversibly along the strand-transfer pathway in a DNA supercoiling-dependent manner, and that the resulting STC is exceptionally stable. We further show how auxiliary DNA-binding interfaces enable dynamic target site selection and undergo conformational changes on strand transfer, suggesting a mechanism for engaging cellular machines involved in STC resolution and repair.

## Results

### Supercoiling drives efficient formation of long-lived STCs.
To study PFV integration, we assembled CI intasomes from purified integrase and mimetics of 3′ processed viral DNA ends. Using supercoiled plasmid DNA as a target (Fig. 1b, c), we can distinguish unreacted molecules (supercoiled) and half-site (open circular) or full-site (linear) reaction products in both AFM[20] and gel images (Supplementary Fig. 1). Samples prepared by incubating pBR322 plasmids with 10 nM wild-type (WT) CI in reaction buffer and direct deposition for AFM imaging resulted in ~25% intasome-bound plasmids overall (Fig. 1d, e). After 4 h incubation, the fraction of bare open circular plasmids increased slightly, whereas bare linearized plasmids, the expected product for full-site integration and subsequent STC disassembly, did not appear in the AFM images. Yet, the fraction of intasome-bound plasmids with open circular topologies increased with increasing incubation time (Supplementary Fig. 2), indicative of strand transfer. Indeed, chemical DNA deproteination (see Methods)

prior to AFM imaging reveals that ~85% of the intasome-bound targets reacted, demonstrating that the strand-transfer yield much exceeds 50% as expected for an iso-energetic reaction. We find that ~71% of the complexes were linearized and ~13% converted to the open circular topology, implying that full-site integration is the dominant reaction (Fig. 1f, g).

To complement the AFM data, we incubated supercoiled DNA with CI intasomes assembled on Atto532-labeled viral DNA mimetics and separated reaction products before and after deproteination by gel electrophoresis (Fig. 1h and Supplementary Fig. 1). Reaction products analyzed under native conditions show a marked increase of an open circular form associated with the labeled viral DNA, compared with unreacted plasmid samples (Fig. 1h, right). Chemical deproteination of the reacted sample yields linearized (full-site) products with incorporated viral DNA, at the expense of the open circular fraction, indicating that STCs allow torsional relaxation, while remaining stably bound, in agreement with our AFM data and with results obtained previously for HIV integration[19]. In our gel data, relaxed topoisomers that would be expected for a reversible reaction are not detected, suggesting that strand transfer is essentially irreversible. Notably, strand transfer in supercoiled targets features an ~2-fold higher yield as compared with topologically relaxed targets (Fig. 1i), despite similar levels of TCCs seen in AFM data ($25 \pm 5\%$ and $23 \pm 5\%$ formed on relaxed and supercoiled plasmids, respectively; errors reflect $\sqrt{n}/n_{tot}$). The twofold lower strand-transfer yield for relaxed vs. supercoiled plasmids suggest an integration yield $\lesssim 50\%$ for relaxed DNA, close to the expectation of an iso-energetic reaction, indicating that the release of supercoiling free energy (estimated in Methods section) drives the integration reaction forward.

### DNA binding at auxiliary interfaces governs target capture.
We next used AFM to probe the geometry of CI, TCC, and STC intasomes. High-resolution topographs revealed intasomes with ellipsoidal shapes (Fig. 2a) and a long axis of $19 \pm 2$ nm (mean ± SD of the full width at half maximum (FWHM) height; $n = 82$), in quantitative agreement with the dimensions obtained via small angle X-ray scattering of PFV intasomes in solution (Fig. 2b, see Methods)[21].

Brief (30 s) incubation of CI with linear target DNA predominantly yields complexes bound to a single target DNA segment (Fig. 2c). Binding occurs in distinct geometries: apical binding with the DNA across the short axis of the ellipsoid near the outer integrase monomers and longitudinal binding with the DNA along the long axis of the intasome (Fig. 2c, d and Supplementary Fig. 3). Surprisingly, exclusive binding to the active site, which runs across the short axis through the intasome's two-dimensional (2D)-projected center of mass, was not observed. K168E mutant intasomes with reduced positive charge at the secondary grooves of the outer integrase dimer interfaces exhibited a ~3-fold reduced affinity for apical target binding (Fig. 2c, d and Supplementary Fig. 3), suggesting that the apical binding interfaces involve the secondary grooves, which were previously found to be critical for in vivo infectivity[22].

Using supercoiled DNA as a target, AFM images revealed WT intasomes predominantly interacting with several DNA strands, indicating that multiple auxiliary interfaces must be engaged (Fig. 2e, f and Supplementary Fig. 3). Interestingly, although complexes bound to three target DNA segments (referred to as branched complexes) occur transiently at short incubation times, longer incubation predominantly yields complexes bound to two target segments (termed bridging complexes; Fig. 2e, f and Supplementary Fig. 3). Control AFM measurements with

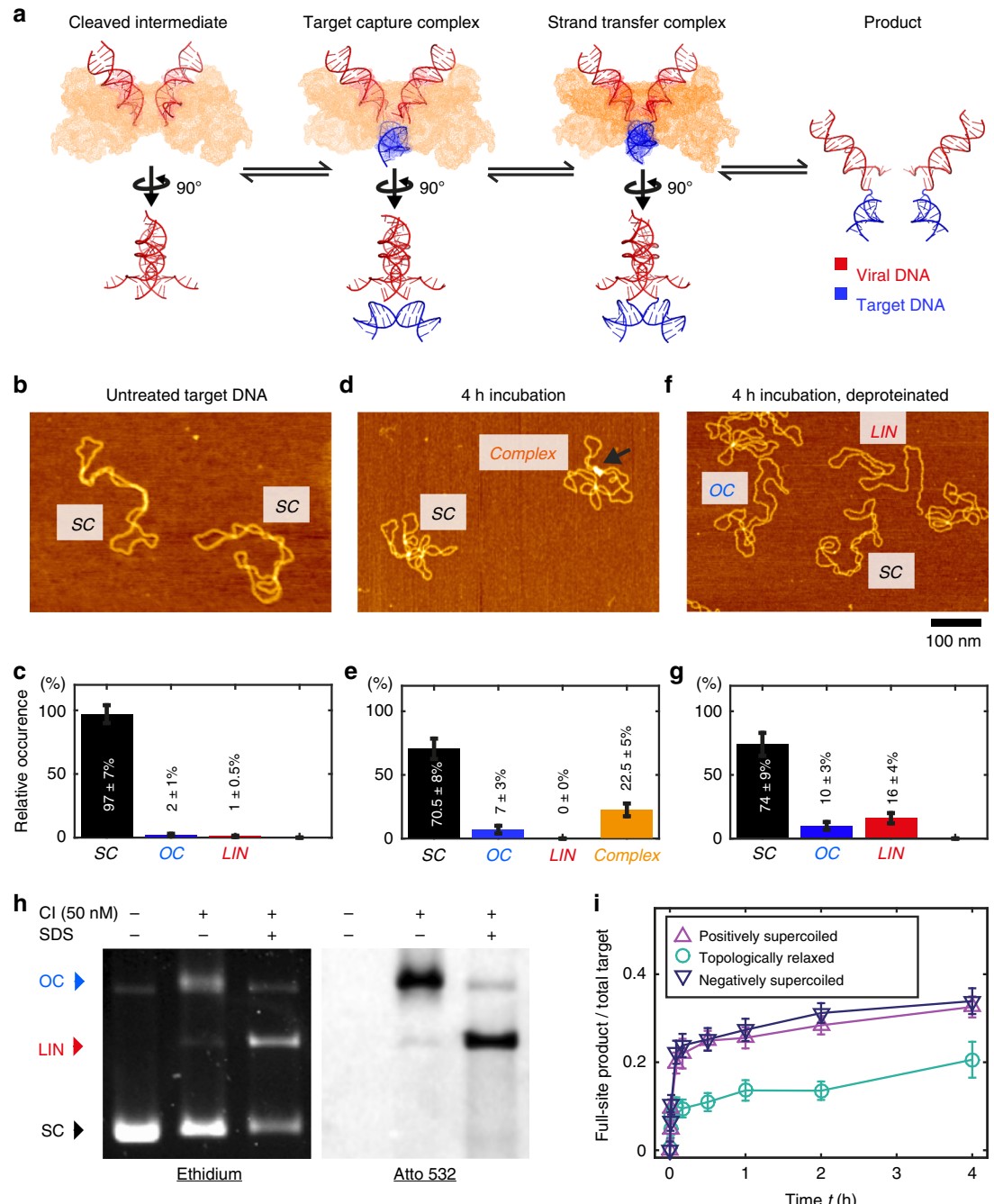

**Fig. 1** Topology and yield of strand transfer intermediates. **a** Reaction schematic of PFV strand transfer with crystal structures of nucleoprotein intermediates (PDB 3l2r, 3os0, 3os1). **b** AFM image of untreated supercoiled plasmids. **c** Relative occurrence of supercoiled (SC), open circular (OC), and linear (LIN) topologies in untreated plasmid samples ($n_{tot} = 214$; errors are $\sqrt{n}/n_{tot}$). **d** AFM image of supercoiled plasmids incubated with CI (10 nM; 4 h, 37 °C). **e** Relative occurrence of different free DNA topologies and nucleoprotein complexes, in samples of supercoiled plasmid incubated with cleaved intermediate (10 nM; 4 h, 37 °C) ($n_{tot} = 518$; errors are $\sqrt{n}/n_{tot}$). **f** AFM image of sample after incubation with CI and subsequent deproteination. **g** Relative occurrence of DNA topological forms in deproteinated samples of supercoiled plasmid incubated with CI ($n_{tot} = 100$; errors are $\sqrt{n}/n_{tot}$). The linearized fraction increases significantly after deproteination compared with the other two conditions ($p < 0.0001$ in both cases). **h** Gel electrophoresis of untreated plasmids (first lane), and native (secondlane) and deproteinated (third lane) reaction products of supercoiled pBR322 and intasomes assembled on Atto532-labeled viral DNA mimetics (left: ethidium bromide stain; right: Atto532 dye). **i** Ratio of full-site product to total DNA as a function of incubation time, deduced from electrophoretic separation of reaction mixtures of plasmids with different supercoiling density and CI intasome (25 nM), followed by Sybr Gold staining. Error bars are SD from two independent repeats. Source data are provided as a Source Data file

intasomes under non-reactive conditions and ensemble kinetic analysis suggest that branched complexes, stabilized by auxiliary DNA-binding interfaces, constitute on-pathway intermediates during target capture (Supplementary Figs. 3 and 4). Further, AFM imaging of gel-purified STCs demonstrate that auxiliary

sites remain engaged in the STC (Fig. 2g and Supplementary Fig. 2).

The exit-angle distributions of DNA in branched and bridging complexes are well-defined, suggesting conserved folding (Fig. 2h, i). Based on high-resolution structural information[14,22] and our

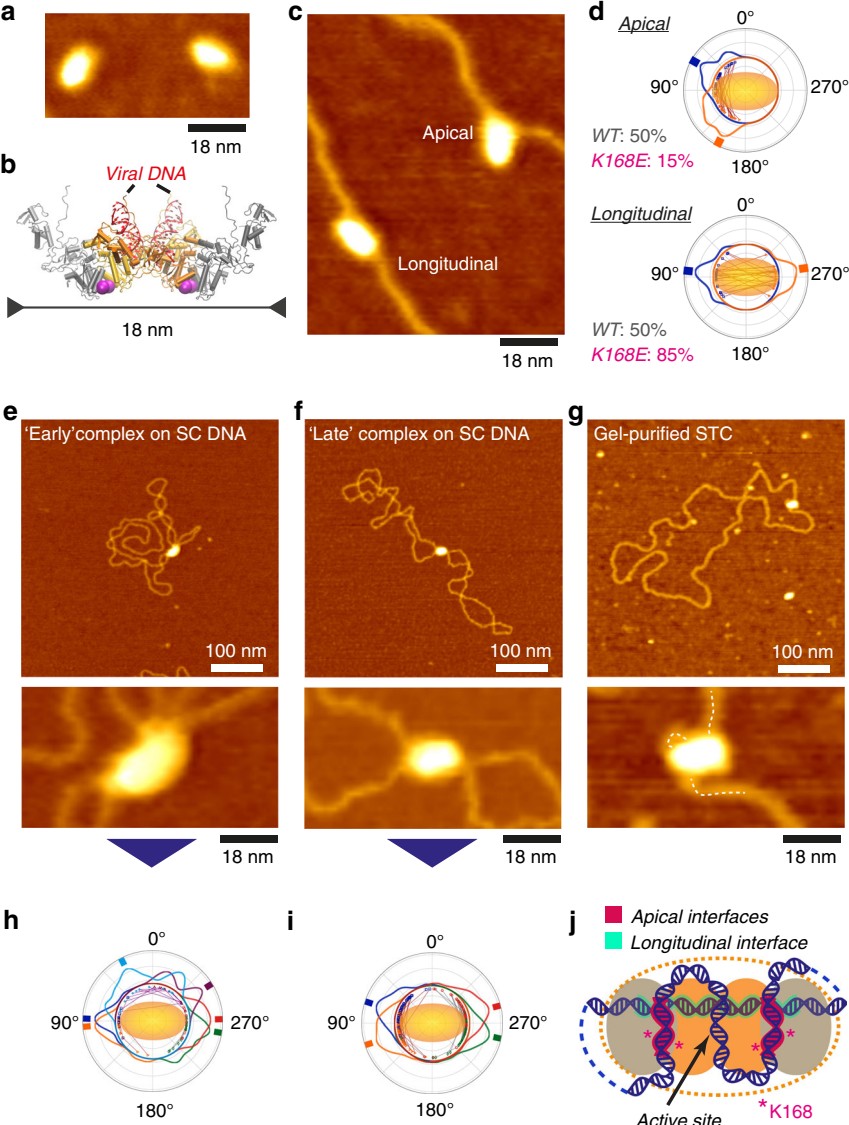

**Fig. 2** Auxiliary DNA-binding interfaces are engaged in TCCs and STCs. **a** AFM images of CI intasomes. **b** Atomistic model of PFV intasome solution structure (*21*). Purple spheres indicate residues K168. Viral DNA mimetics (red) are not visible in our AFM data. **c** Early TCCs formed on linear pBR322 DNA reveal longitudinal and apical binding geometries. **d** Polar plots of entry and exit angles with respect to the intasome long axis for apical and longitudinal binding modes, and relative occurrence in WT and K168E intasomes. **e** AFM image of intasomes incubated briefly (2 min) with supercoiled plasmid DNA, depicting a branched complex as found in ~50% of early complexes. **f** AFM image of a bridging complex that dominates (~80%) the population of complexes at longer (>45 min) incubation. **g** AFM image of a gel-purified STC. **h** Polar plot of exit angles in branched complexes. **i** Polar plot of exit angles in bridging complexes. **j** Model for target DNA (blue) folding in TCC and STC intasomes (yellow dotted contour). Viral DNA mimetics are not shown. Source data are provided as a Source Data file

AFM data, we propose a minimal low-resolution model of the DNA folding in the TCC and STC (Fig. 2j). The ellipsoidal intasome has the active site positioned centrally along the minor axis. The apical auxiliary interfaces correspond to the secondary grooves involving residues K168 located parallel to the left and right of the active site on the same face of the complex (Fig. 2j). The stability and long lifetime of the STC (Fig. 1) combined with the observation that K168E intasomes exhibit faster topological relaxation and complex disassembly after reaction with supercoiled targets as compared with WT intasomes (Supplementary Fig. 2) strongly indicates that the single strand breaks generated at the active site are mechanically shielded by the apical interfaces. To protect the DNA at the active site, the same target DNA segment must bind both apical interfaces, suggesting an S-shaped or wrapped path of the DNA (Fig. 2j). Indeed, AFM images of

purified STCs are consistent with an S-shaped DNA path and the apical interfaces, and the active site remaining bound (Fig. 2g and Supplementary Fig. 2). The paths of the DNA segments entering and exiting the intasome complex require one more DNA segment to be bound (Fig. 2h, i), which we assign to the longitudinal interface (Fig. 2j and Supplementary Fig. 2).

**Real-time observation demonstrates a dynamic target search.** Building on the mesoscale structural data obtained via AFM imaging, we next developed a multiplexed MT assay to probe integration dynamics in real time. Target DNA molecules are tethered between a flow cell surface and magnetic beads, and can be stretched and supercoiled using external magnets[23]. Plectoneme formation upon supercoiling decreases the tether extension

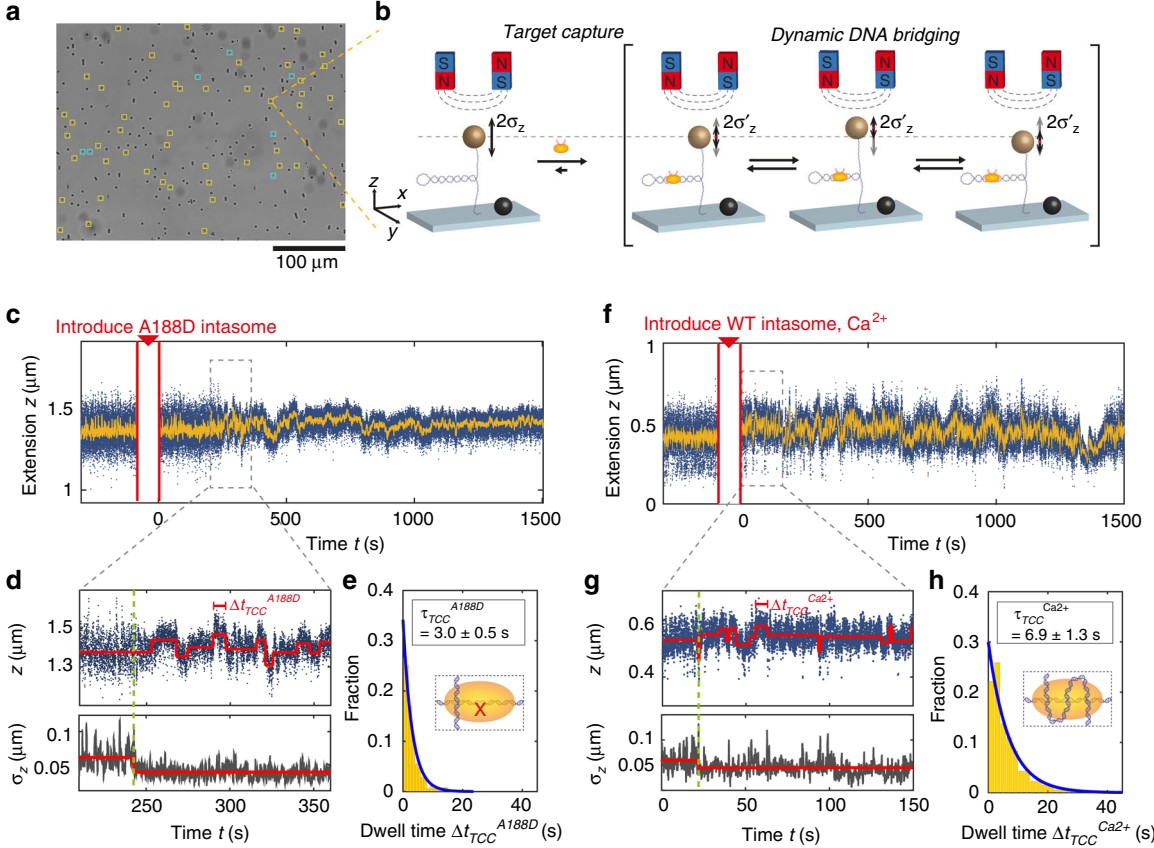

**Fig. 3** Magnetic tweezers assay reveals target capture dynamics. **a** A typical field of view depicting ~50 beads used for tracking (yellow) and reference beads (blue). **b** DNA tethers are supercoiled using external magnets and exhibit extension fluctuations with SD $\sigma_z$. Target capture reduces $\sigma_z$ via DNA bridging. Transient interface unbinding repositions intasomes thereby affecting extension. **c** Time trace of supercoiled target DNA extension before and after binding A188D CI intasomes (blue: raw data at 58 Hz, yellow: 1 Hz smoothed data). **d** Enlarged region of **c** highlighting the onset of dynamic bridging (green dotted line) and coinciding $\sigma_z$ reduction calculated with a 0.5 s moving window (red lines: trace fitted using step-finding algorithm)[24]. **e** Exponential fit of $\Delta t_{TCC}^{A188D}$ distribution yields a lifetime of auxiliary interfaces $\tau_{TCC}^{A188D} = 3.0 \pm 0.5$ s. (Error is 95% CI; $n = 724$). **f** Time trace of supercoiled DNA extension and response to binding WT CI in Ca$^{2+}$ buffer. **g** Enlarged region of **f** shows the onset of dynamic bridging and $\sigma_z$ reduction (green dotted line). **h** Exponential fit of $\Delta t_{TCC}^{Ca2+}$ distribution with lifetime $\tau_{TCC}^{Ca2+} = 6.9 \pm 1.3$ s. (Error is 95% CI; $n = 750$). Source data are provided as a Source Data file

compared with torsionally relaxed molecules. In our assay, typically ~50 magnetic beads tethered by a single double-stranded DNA molecule are selected for tracking per field of view (Fig. 3a). We first record extension traces of supercoiled DNA and then introduce CI intasomes (10 nM) into the flow cell, which results in ~30% of DNA tethers (296 out of 934 tethers overall) exhibiting signatures in their extension that deviate from the bare DNA behavior. Consequently, we expect that tether interactions predominantly (~90%) involve a single intasome, in agreement with our AFM observations.

We first investigated intasomes under conditions that allow target capture yet inhibit strand transfer, by using either intasomes carrying the A188D mutation that blocks the active site, by adding the strand-transfer inhibitor Raltegravir, or by using Ca$^{2+}$ instead of Mg$^{2+}$ in the reaction buffer (Fig. 3). In all cases, we found that intasome binding reduced the level of extension fluctuations $\sigma_z$ (Fig. 3d, g), whereas at the same time discrete transitions in extension occur (Fig. 3c, d, f, g). The reduction of $\sigma_z$ is consistent with a reduction in effective contour length by pinching off a segment of the plectoneme (Supplementary Fig. 5), in line with our observation that target capture bridges DNA segments by multivalent binding. As binding and bridging can occur at any position along the plectoneme, the reduction of $\sigma_z$ is variable (Supplementary Fig. 5). To quantify the dynamics of the stepping behavior, we fitted the traces using a

step-finding algorithm[24] and quantified step sizes and dwell times between steps (Fig. 3d, g). The step-size distribution is continuous and decays exponentially (decay length ~45 nm), consistent with interface unbinding and rapid rebinding of a nearby target DNA segment (Supplementary Fig. 6).

The distributions of dwell times in all cases are well described by single exponentials. In experiments with intasomes with blocked active site (either via the A188D mutation or addition of Raltegravir), the lifetimes determined from the exponential fits is $\tau_{A188D}^{TCC} = 3.0 \pm 0.2$ s (error is 95% CI; Fig. 3c-e), or $\tau_{Ralt}^{TCC} = 2.7 \pm 0.2$ s (error is 95% CI; Supplementary Fig. 6). These lifetimes reflect the timescale of association between target DNA and auxiliary binding sites in TCCs, and is in good agreement with lifetimes on linear DNA found by single-particle tracking experiments[16]. Experiments performed with WT intasomes using a buffer containing Ca$^{2+}$ and no Mg$^{2+}$, to allow active site binding but suppress catalysis, found a lifetime $\tau_{TCC}^{Ca2+} = 6.9 \pm 1.3$ s (error is 95% CI; Fig. 3f-h), which is ~2-fold larger than for A188D intasomes or the Raltegravir condition, suggesting that although active site binding contributes to the lifetime of the interaction, it is short lived.

The dynamic changes in tether extension $z$ and the reduction in $\sigma_z$ after target capture continue for hours, and we only very rarely (1 out of 76 traces; corresponding to ~200 h of total observation time) see a return to the behavior of bare DNA that is

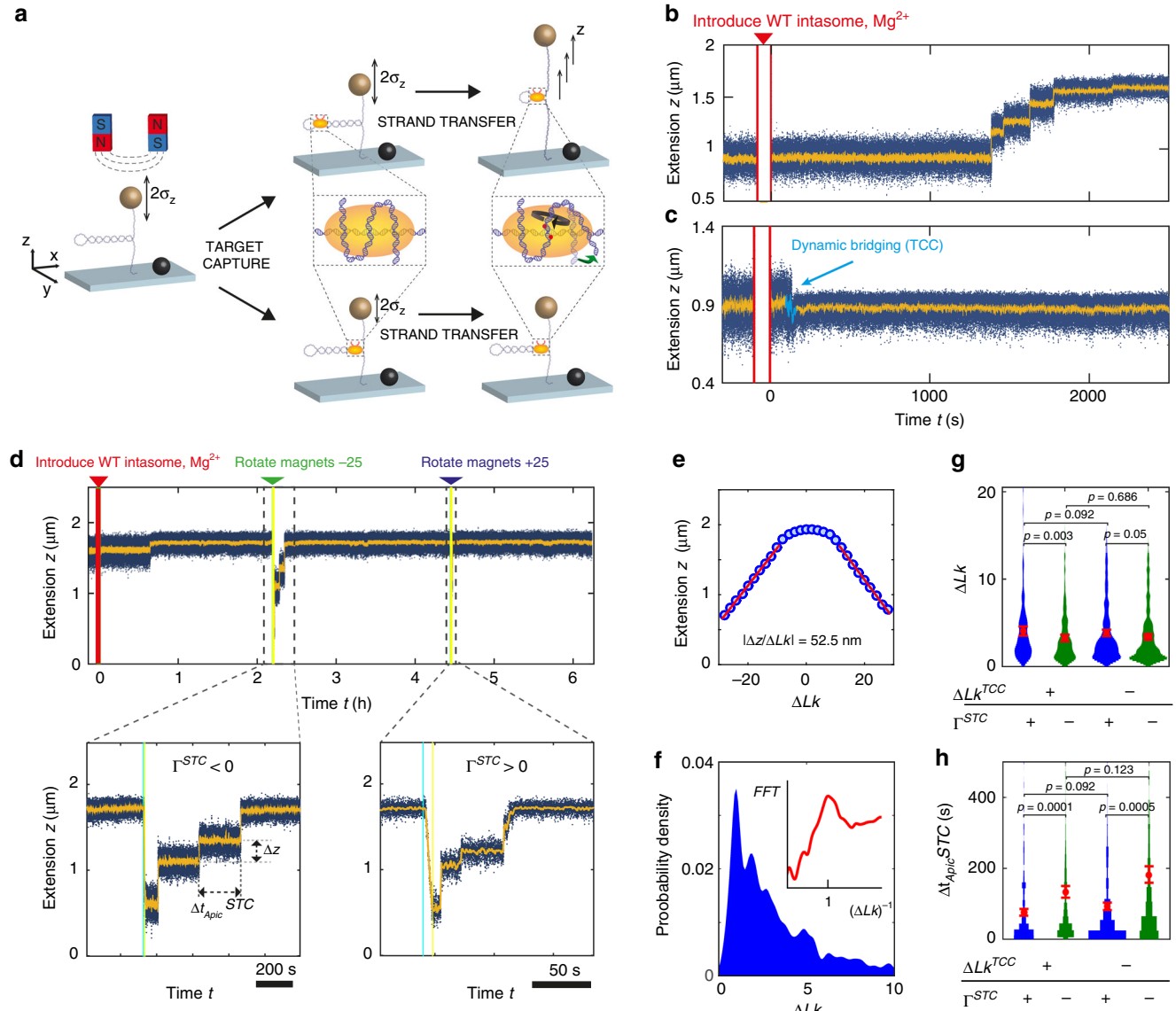

**Fig. 4** Real-time observation of strand transfer and apical interface stability in STCs. **a** Scheme depicting signatures of strand transfer. Top: binding and reaction near plectoneme end loops minimally affect $\sigma_z$ but enables supercoil release through apical interface unbinding (green arrow) and rotational relaxation (black arrow). Bottom: binding near the origin of the plectoneme suppresses $\sigma_z$, strand transfer quenches dynamic bridging by intasome-target anchoring. **b**, **c** Extension time-traces of supercoiled target DNA reacting with intasome, depicting stepwise extension increments (**b**) and extension hopping followed by a stable extension level (**c**). **d** Extension time-trace of supercoiled target DNA reacting with WT intasome. External magnets introduce supercoils that are released in steps due to transient apical interface unbinding. Extension plateaus quantify dwell times $\Delta t_{Apic}^{STC}$, step sizes $\Delta z$ quantify extension increments. **e** In the plectonemic regime, the number of turns released per unbinding event $\Delta Lk$ is proportional to $\Delta z$. **f** $\Delta Lk$ distribution (kernel density estimate; bandwidth 0.2 turns). Inset: Fourier transformation after subtracting an exponential background. **g** Dependence of step size $\Delta Lk$ distribution on the sign of $\Delta Lk^{TCC}$ in TCCs and on the sign of the torque $\Gamma^{STC}$ applied to STCs (red data points are mean $\langle\Delta Lk\rangle$ and error bars are 95% CI as obtained from an exponential fit; Supplementary Fig. 7). **h** Dependence of dwell times $\Delta t_{Apic}^{STC}$ on $\Delta Lk^{TCC}$ and $\Gamma^{STC}$ (red data points are mean lifetime $\tau_{Apic}^{STC}$ and error bars are 95% CI as obtained from an exponential fit; Supplementary Fig. 7). Significance calculated using two-sample Kolmogorov–Smirnov test ($n^{TCC+}_{STC+} = 227$; $n^{TCC+}_{STC-} = 288$; $n^{TCC-}_{STC+} = 412$; $n^{TCC-}_{STC-} = 444$). Source data are provided as a Source Data file

indicative of complete intasome dissociation. Thus, target capture is virtually irreversible owing to multivalent binding at auxiliary interfaces. The ~ seconds lifetime of binding to (at least) one auxiliary interface and the active site allows dynamic integration site selection, while preserving continuous contact between intasome and target DNA for extended periods of time.

**Strand transfer stabilizes apical auxiliary interfaces**. On introducing catalytically competent intasomes, the dynamic and rapid changes in tether extension level, seen consistently with

catalytically inactive intasomes (Fig. 3), are only seen transiently. Instead, we see two additional signatures in the MT extension traces (Fig. 4a-c). A first signature is rapid stepwise increases in extension beyond the initial range of fluctuations that we interpret as supercoil release following strand transfer-induced generation of single strand breaks (Fig. 4a, b). The stepwise pattern is explained by our model for the STC (Fig. 2i), wherein apical interfaces shield the breaks (Supplementary Fig. 2) and supercoils can only relax on transient unbinding. A second signature is sampling of different extension states followed by a stable extension level, consistent with covalent anchoring of the target

DNA to the intasome on strand transfer that stops the dynamic target search (Fig. 4a, c). Establishing a stable extension level is typically fast (<1 min; Fig. 4c and Supplementary Fig. 6) and irreversible in all observations.

The extent of fluctuation reduction observed in the traces is anti-correlated with the total extension increase by stepwise supercoil release, which can be understood from the variable positions of binding and reaction along the plectoneme (Supplementary Fig. 5). DNA binding close to the plectoneme end loop allows large extension increments via supercoil release but features minimal $\sigma_z$ reduction (Fig. 4a, top, and Supplementary Fig. 5). Conversely, binding near the plectoneme origin significantly reduces extension fluctuations $\sigma_z$ but allows relaxation of few or no supercoils (Fig. 4a, bottom, and Supplementary Fig. 5).

We can controllably introduce new supercoils by rotating the magnets. For reacted DNA tethers, new supercoils are released in a stepwise manner, independent of the position of intasome binding and reaction (Fig. 4d). The time-traces of stepwise supercoil relaxation provide information on apical interface binding kinetics. We exploit the linear relation between extension and linking difference in the plectonemic regime (Fig. 4e) to convert step heights $\Delta z$ to number of supercoils $\Delta Lk$ released per dissociation event[25,26]. The probability distribution of $\Delta Lk$ exhibits maxima at integer values (Fig. 4f), implying that apical interfaces involve specific contacts with the target DNA. The number of supercoils removed per event is exponentially distributed, indicative of a mechanism where the interface has a fixed probability of reforming every turn[25]. Large supercoil relaxation steps ($\Delta Lk > 10$) occur significantly more frequently for K168E intasomes compared with WT ($p < 0.0001$; two-sample $t$-test; Supplementary Fig. 7), highlighting the stabilizing contribution of the secondary grooves at the outer dimer interface.

We evaluated whether stepwise supercoil removal is affected by (i) the sign of DNA supercoiling on which the TCC is assembled $\Delta Lk^{TCC}$ and (ii) the sign of the supercoils introduced by the external magnets after strand transfer (and consequently the torque on the STC $\Gamma^{STC}$). We find that the rebinding probability is independent of $\Delta Lk^{TCC}$ (Fig. 4g and Supplementary Fig. 7), both for the removal of positive and negative supercoils. In contrast, a larger mean number of supercoils removed per step $\langle \Delta Lk \rangle$ is observed under positive compared with negative $\Gamma^{STC}$ (Fig. 4g and Supplementary Fig. 7). Furthermore, relaxation traces allow quantification of the dwell times of individual plateaus, which report on the apical interface lifetime $\Delta t_{apic}^{STC}$. Dwell times are exponentially distributed in all cases (Supplementary Fig. 7). The effect of supercoil chirality follows a pattern similar to the step size distributions: there is no significant difference in lifetime $\tau_{Apic}^{STC}$ between TCCs assembled on positively or negatively supercoiled target DNA for the removal of supercoils of either sign (Fig. 5h). In contrast, there are significant differences in lifetime for removal of positive supercoils compared with negative supercoils. Larger average step sizes, corresponding to a lower rebinding probability per turn, and shorter average dwell times for the removal of positive compared with negative supercoils both suggest that a positive torque $\Gamma^{STC}$ destabilizes the apical interface more than a negative torque. Overall, the apical interface lifetime after reaction in the STCs ($\tau_{Apic}^{STC} \sim 90$ s, depending on the applied torque $\Gamma^{STC}$) is at least 20-fold longer than the lifetime of the auxiliary interfaces before reaction in the TCC ($\tau_{A188D}^{TCC} = 3.0 \pm 0.5$ s; error is 95% CI) implying stabilization of target DNA–intasome interactions beyond the active site after strand transfer.

**Forced disassembly of the STC.** Remarkably, in our MT assay at low forces ($F = 0.5$ pN; 223 reacted tethers), we never observed

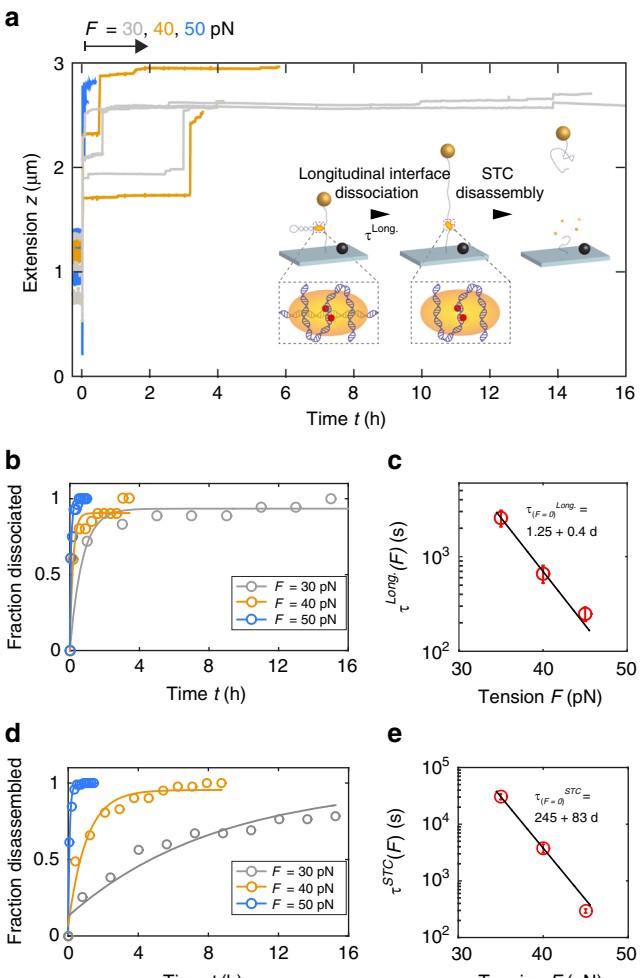

**Fig. 5** Forced disassembly of STCs. **a** Typical extension time traces of reacted tethers subjected to high tension (30, 40, 50 pN) depicting large extension jumps preceding bead release ($n^{30pN} = 55$; $n^{40pN} = 41$; $n^{50pN} = 36$). **b** Cumulative distributions of dissociated longitudinal interfaces over time for forces 30, 40, 50 pN, and exponential fits to the data yielding lifetimes $\tau_{Long.}(F)$. **c** Dependence of lifetime $\tau_{Long.}$ on tension $F$ and fit to the Bell model ($\tau_{(F=0)}^{Long.} = 1.25 \pm 0.$ d $\Delta x^{Long.} = 5.1 \pm 0.5$ Å; errors are standard fit error). **d** Cumulative distributions of disassembled STCs over time for forces 30, 40, 50 pN and exponential fits to the data yielding lifetimes $\tau_{STC}(F)$. **e** Dependence of lifetimes $\tau_{STC}$ on tension $F$ and fit to the Bell model ($\tau_{(F=0)}^{STC} = 245 \pm 83$ d; $\Delta x^{STC} = 8.9 \pm 0.4$ Å; errors are standard fit error). Source data are provided as a Source Data file

release of the DNA-tethered beads after reaction (Fig. 4), which would be expected for disassembly of the STC after completion of the integration reaction. The high stability of the STC in the MT is consistent with the results from AFM imaging (Fig. 1e) and gel electrophoresis (Fig. 1i), which also suggest a very stable and long-lived STC. To directly quantify the mechanical stability of the STC, we use the MT's ability to apply large and precisely calibrated stretching forces and subject STCs to varying levels of tension. Upon applying high tension ($F \geq 30$ pN), tether extensions immediately increase due to elastic DNA stretching and removal of plectonemes, but beads remain tethered for ~hours before the eventual bead release that we interpret as STC disassembly (Fig. 5a). Interestingly, tethers that exhibited strongly reduced extension fluctuations at low force, indicative of STC formation near the plectoneme origin, initially show increased extension on increasing tension to values much smaller than the expected DNA contour

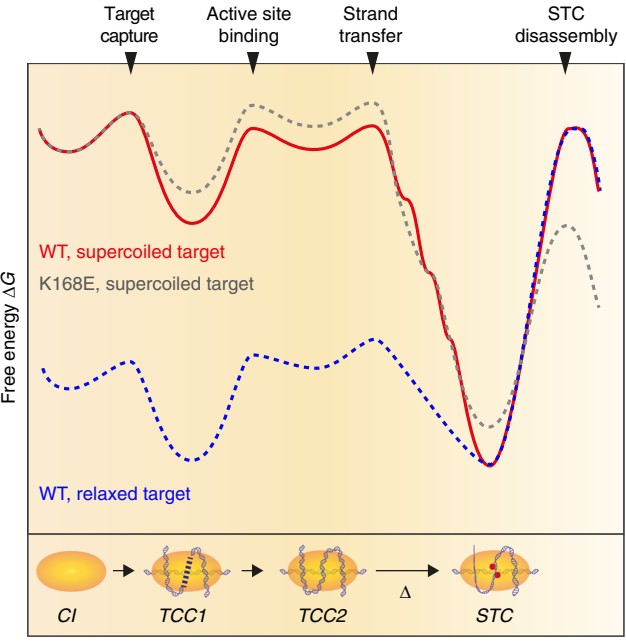

**Fig. 6** Model of the integration energy landscape. Target capture occurs in two stages and is virtually irreversible yet dynamic, owing to multivalent binding at auxiliary interfaces that form independent of the sign of target DNA supercoiling. Active site binding requires severe DNA deformation and is rate-limiting with respect to strand transfer. Strand transfer is irreversible due to stepwise release of mechanical strain in the target and results in an allosteric conformational change (Δ) that stabilizes auxiliary interfaces in a torque-dependent manner. Target binding at apical interfaces mechanically shields the DNA in the active site and renders STC disassembly extremely slow

length. In ~90% of those cases, an extension jump precedes bead release. The size of extension jumps correlates strongly with the extent of z-fluctuation reduction $\Delta\sigma_z$ (Fig. S5). We therefore interpret extension jumps preceeding bead release as the result of loop release. Our structural model suggests that the sequestered DNA loop can be released only by dissociation of the longitudinal interface (Fig. 2j). Survival times for both loop release and the final STC disassembly under force are exponentially distributed (Fig. 5b, d). The fitted lifetimes $\tau_{(F)}$ exhibit an exponential force dependence[27] $\tau_{(F)} = \tau_{(F=0)} \cdot \exp\left(\frac{-F \cdot \Delta x}{k_B T}\right)$, with zero-force lifetime $\tau_{(F=0)}^{STC} = 245 \pm 83$ d (standard fit error) for the STC (Fig. 5e) and $\tau_{(F=0)}^{Long.} = 1.25 \pm 0.5$ d (standard fit error) for longitudinal interface unbinding (Fig. 5c), demonstrating the extreme stability of the STC towards disassembly.

## Discussion

Using single-molecule assays, we have directly probed the PFV integration landscape (Fig. 6) in real time. Our data reveal that target capture and engagement of the active site is the rate-limiting step of integration (Supplementary Fig. 4) caused by the requirement for strong target deformation, yet irreversible, owing to multivalent interactions at auxiliary interfaces that collectively warrant a high affinity for the target (Fig. 3). Importantly, individual interactions with the target at auxiliary interfaces and the active site are sufficiently weak to enable dynamic site selection. Nevertheless, on supercoiled targets, site selection is limited and quickly followed by strand transfer (Supplementary Fig. 6).

The chemistry of strand transfer is intrinsically iso-energetic, yet our data indicate a high (~85%) yield when the target is

supercoiled (Fig. 1), in stark contrast with intasomes bound to topologically relaxed targets that achieve ≲50% conversion (Fig. 1i). Mutant intasomes with significantly lower binding energy react with yields similar to the WT (Supplementary Fig. 2). Accordingly, we propose that the release of supercoiling free energy (~100 $k_B T$; Methods) following the generation of single-strand breaks on strand transfer is the main driving force for shifting the transesterification equilibrium forward.

After strand transfer, STCs do not disassemble spontaneously (Figs. 1 and 4). We used the ability of MT to apply high tension and to quantitatively challenge STC stability. STC lifetimes were found to decrease exponentially with applied force and extrapolation yields a zero-force lifetime of ~8 months (Fig. 5), much longer than the retroviral replication cycle (~1 day). As STC resolution is the key towards establishing a functional provirus, our data strongly suggest the necessity of enzymatic STC resolution in vivo. Interestingly, the lifetime of auxiliary DNA-binding interfaces is significantly increased in STCs compared with TCCs. In addition, the stability of apical interfaces in STCs is ~2-fold higher under negative as compared with positive torque, whereas initial supercoil chirality of the target does not affect target capture (Fig. 4). Taken together, our data imply that strand transfer triggers chiral conformational changes that stabilize and rearrange the intasome at apical interfaces, i.e., well beyond the active site. We hypothesize that these conformational changes might signal strand transfer completion to engage cellular machinery involved in STC resolution and repair.

The finding that PFV intasomes employ auxiliary-binding sites for modulating the barriers to integration raises the question how the topology of higher-order intasomes governs integration of pathogenic retroviruses, most notably HIV. The single-molecule assays developed in this work are expected to be particularly useful to further unravel the complexity of this important class of molecular machines.

## Methods

**Integrase expression and purification**. BL21 *Escherichia coli* cells (Agilent Technologies) were grown to OD$_{600}$ 0.9–1.0 at 29 °C prior to the addition of 0.25 mM IPTG and 50 μM ZnCl$_2$ for 4 h at 25 °C. Cells were collected and stored at −80 °C. Protein purification started with the resuspending the BL21 *E. coli* cells in 25 mL HSB buffer (50 mM Tris-HCl pH 7.4, 500 mM NaCl). Afterwards, 0.5 mM phenylmethylsulfonyl fluoride, 1 U/10 mL DNase, and 1 mg/mL lysozyme were added, and the mixture was incubated for 10–15 min before sonication (6 × 30 s or until homogenized). The lysate was cleared by centrifugation for 45 min at 18,000 r.p.m. Before loading the lysate on the Ni-Sepharose beads, the beads were washed and HSB supplemented with 20 mM imidazole. The cleared lysate was incubated for 30 min at 4 °C with Ni-Sepharose. After lysis flow through, the beads were washed with five-column volumes and HSB supplemented with 20 mM imidazole. Protein was eluted with HSB supplemented with 200 mM imidazole and 10 mM dithiothreitol (DTT). The concentration of the eluted protein was determined by measuring the absorbance at 280 nm ($\varepsilon_{280} = 58{,}110$ M$^{-1}$ cm$^{-1}$, MW = 47,485 Da). The fractions containing PFV integrase were pooled and digest overnight at 4 °C with HRV3C protease during dialysis to HSB supplemented with 10 mM DTT (1:100 w/w ratio of protease: integrase). Afterwards, the protein was cleared from the His-tag by the use of a Ni$^{2+}$ and additionally a GSH column. The protein was dialyzed against HSB supplemented with 5 mM DTT and 10% glycerol.

**PFV intasome assembly and purification**. Intasome assembly and purification was performed according to published protocols[10]. Briefly, PFV integrase (120 μM) was mixed with synthetic DNA duplex (5′-TGCGAAATTCCATGACA-3′ and 5′-ATTGTCATGGAATTTCGCA-3′; 50 μM) in 50 mM BisTris propane-HCl buffer supplemented with 500 mM NaCl (pH 7.45; 500 μL). For assembly of fluorescently labeled intasomes, we used the modified oligo 5′-/5ATTO532N/TGC GAA ATT CCA TGA CA-3′. The mixture was then dialyzed for 18 h at 18 °C against 200 mM NaCl, 2 mM DTT, 25 μM ZnCl$_2$, 20 mM BisTris propane-HCl, pH 7.45. Next, a further dialysis step for 3 h to the dialysate was performed. Finally, 120 mM NaCl were added and the sample was incubated on ice for 1 h. Size-exclusion gel chromatography (Superdex 200 10/300 GL column) coupled to an ÄKTA Purifier system (GE Healthcare) was employed to fractionate the assembly, with mobile phase (0.32 M NaCl, 20 mM BisTris propane-HCl, pH 7.45) at 1 mL/min, 4 °C. The concentration of the intasome was determined spectrometrically using molar extinction coefficients $\varepsilon_{280} = 626{,}329$ M$^{-1}$ cm$^{-1}$ and $\varepsilon_{260} = 841{,}491$ M$^{-1}$ cm$^{-1}$.

**DNA plasmids for AFM and bulk integration assays**. Negatively supercoiled circular DNA plasmids (pUC19: 2686 bp, pBR322: 4361 bp, M13mp18 RF I DNA: 7249 bp) were obtained commercially (New England Biolabs, Ipswich, MA, USA). To construct the 1800 bp plasmid, commercial pUC19 plasmid (New England Biolabs, Ipswich, MA, USA) was linearized with two primers: FW pUC19 (EcoRV) (5′-GATATCCGTAAAAAGGCCGCG-3′) and REV pUC19 (BspQI) (5′-GCTC TTCCCTTAGACGTCAGGTGGC-3′). The linearization reaction was carried out via PCR reaction with a Phusion master mix (2× Phusion High-Fidelity PCR Master Mix, Thermo Fisher Scientific, Inc., Waltham, MA, USA). One nanogram DNA was used as template for the PCR reaction. The PCR program was: 98 °C 2 min, 30 × (98 °C 10 s, 55 °C 10 s, 72 °C 36 s), 72 °C 5 min. After the PCR reaction was completed, the product was analyzed on a 1% (w/v) agarose gel. The PCR product was directly used in a ligation reaction supplemented with DpnI and T4 Polynucleotide Kinase (PNK; New England Biolabs, Ipswich, MA, USA). DpnI was used to remove all initial DNA template and T4 PNK was used to phosphorylate the DNA for re-ligation. The ligation reaction (1 µl T4 Ligase, 1 µl ATP (10 mM), 0.5 µl PEG-6000, 1 µl T4 PNK, 1 µl DpnI, and 5.5 µl PCR reaction mixture), buffered in CutSmart buffer, was incubated for 15 min at 37 °C, continued with 22 °C for 45 min. One microliter of the reaction was used to transform E. coli DH5alpha cells (New England Biolabs). Few clones were picked for overnight cultures to isolate their plasmid (QIAprep Spin Miniprep Kit, Qiagen, Hilden, Germany). Sequence analysis of the full plasmid confirmed its sequence.

Topologically relaxed pBR322 was generated from commercially available, negatively supercoiled pBR322, by treatment with Wheat Germ Topoisomerase I (Inspiralis, Ipswich, UK), in the PFV reaction buffer at 37 °C. The reaction product was purified using phenol–chloroform extraction and ethanol precipitation. The pellet was resuspended in Tris-HCl buffer (10 mM, pH 8.0). Positively supercoiled pBR322 was generated starting from topologically relaxed pBR322 (see above) employing the archaeal histone-like protein rHmfb[20]. Under conditions of low-to-intermediate ionic strength, rHmfb wraps DNA in a right-handed toroidal manner. Topoisomerization and subsequent protein removal results in positively supercoiled plasmid DNA. We followed the protocol for production of rHmfb and its use in the generation of positively supercoiled DNA, as originally reported by LaMarr et al.[28] Briefly, relaxed pBR322 (500 ng) and rHmfb (250 ng) were incubated for 20 min at 37 °C, in a 25 µL volume of aqueous buffer (10 mM Tris-HCl pH 8.0; 2 mM $K_2HPO_4$; 1 mM EDTA; 50 mM NaCl). Next, 2.7 µL containing 3 units of Wheat Germ Topoisomerase I (Inspiralis, Ipswich UK) in buffer (295 mM Tris-HCl, pH 8.0; 4 mM $H_2KPO_4$; 11 mM EDTA; 50 mM NaCl) was added to the solution. Topoisomerization at 37 °C was performed for 90 min. Phenol–chloroform extraction and ethanol precipitation were used to purify the reaction products.

**AFM strand transfer assay**. Samples were prepared as described in previously published protocols[29]. Briefly, we mixed intasome and pBR322 plasmid DNA in reaction buffer (125 mM NaOAc, 5 mM Mg(OAc)₂, 25 mM Tris-HCl (pH 7.4), 10 µM ZnCl₂, 1 mM DTT) in a volume of 100 µL at final concentrations of 10 nM and 0.5 ng/µL, respectively. After incubation at 37 °C, 20 µL of the sample was drop-casted on poly-L-lysine (0.01% w/v)-coated muscovite mica (West Chester, USA). For analysis of the DNA topology after reaction, we applied the additional steps of deproteinating the reaction mixture using a PCR purification kit (Zymo Research) followed by adjusting the salt concentration of the eluate to 125 mM NaOAc, 5 mM Mg(OAc)₂, 25 mM Tris-HCl (pH 7.4), 10 µM ZnCl₂. After 30 s, the substrates were gently rinsed using 20 mL of milliQ water and dried using a gentle stream of filtered N₂ gas. AFM imaging was performed on a commercial Multi-mode AFM, equipped with a Nanoscope III controller and a type E scanner. Images were recorded in amplitude modulation mode under ambient conditions and by using silicon cantilevers (Nanoworld; type SSS-NCH; resonance frequency ≈300 kHz; typical end-radius 2 nm; half-cone angle <10 deg). Typical scans were recorded at 1–3 Hz line frequency, with optimized feedback parameters and at 512 × 512 pixels. For image processing and analysis, Scanning Probe Image Processor (SPIP v6.4; Image Metrology) was employed. Image processing involved background correction using global fitting with a third-order polynomial and line-by-line correction through the histogram alignment routine.

**AFM data analysis**. Assignment of DNA topology (open circular vs. supercoiled) was performed by counting the number of self-crossings in circular DNA reports on its topological state[30]. Under our conditions, open circular DNA exhibits a distribution of the number of self-crossings n characterized by a mean $\langle n \rangle = 2.4$ and SD $\sigma_n = 1.7$ (Supplementary Fig. S1). In contrast, negatively supercoiled plasmids of the same length are characterized by a distribution with $\langle n \rangle = 12.2$ and SD $\sigma_n = 1.7$ (Supplementary Fig. S1). At the level of single molecules, we assign the open circular topology for $n \leq 6$ and the covalently closed topology when $n > 6$.

In our topographic images, intasomes are readily distinguishable from simple DNA crossovers or branch points based on the maximum height, which is significantly larger for the intasome (>3.5 nm) as compared with a branch point (~2 nm). Analysis of the DNA entry and exit angles at intasomes was done by quantifying the angle relative to the long axis of the intasome and by connecting the entry/exit point of the DNA, with the center of mass of the intasome. In a first step, the long axis of each intasome was determined using the particle analysis toolbox in SPIP (v.6.4, Image Metrology, Hørsholm, Denmark) and the

nucleoprotein complex was rotated over the smallest angle required to align the long axis of the intasome to the horizontal. Next, the 2D-projected center of mass was determined. Entry and exit angles of DNA loops were determined relative to the long axis, and with the intasome center of mass as the origin.

We use the FWHM to quantify the dimensions along the long axis of intasomes. Intasomes have a measured height $h \sim 4$ nm and actual dimension $D = 18$ nm along the long axis (as deduced from small-angle X-ray scattering studies). A simple geometrical model[31] based on the half-cone angle $\alpha \sim 10$ deg of the AFM tip predicts FWHM $= D + h.\tan(\alpha) = 18.7$ nm, very close to the actual dimension $D$ and in quantitative agreement with the measured FWHM $= 19 \pm 2$ nm.

**Native gel electrophoresis and purification of STCs**. To selectively investigate STCs by AFM, we incubated 200 ng of negatively supercoiled pBR322 plasmid with 10 nM of PFV intasome for 4 h in reaction buffer (volume = 50 µL), at 37 °C. Next, the reacted sample was mixed with loading dye (20 mM Tris, 20 mM acetic acid, 15% (w/v) Ficoll-400, Orange G), and loaded into a broad well on a 1% agarose gel. After separation by electrophoresis for 2 h (75 V) in Tris-acetate buffer, a small section of the gel was cut and removed for staining with Sybr Gold for 30 min. Examples of uncropped and unprocessed scans of a native gel are supplied in the Source Data file. The position of the band comprising the STC (~open circular) is marked under UV irradiation. Next, the excised portion of the gel, marked for the position of the STC, was carefully realigned with the unstained portion of the gel. At the position of the STC, a thin slice of the gel was excised. Gentle squeezing between two glass slides (covered with parafilm) removes the aqueous solution contained within the gel. This solution is mixed 1:1 with high salt buffer (400 mM NaOAc, 25 mM Tris-HCl, pH 7.4) and is directly drop-casted on poly-L-lysine-coated mica for AFM imaging.

**Bulk strand-transfer assays**. Bulk integration kinetic assays were carried out by mixing plasmid DNA with CI (25 nM for the topology dependence; 10 nM for the length dependence), in reaction buffer (125 mM NaOAc, 5 mM Mg(OAc)₂, 25 mM Tris-HCl (pH 7.4), 10 µM ZnCl₂, 1 mM DTT) at 37 °C. Importantly, under the conditions used we find maximally ~40% conversion of the supercoiled plasmid, which ensures a minimal number of targets to react with more than one intasome. Aliquots of 12 µL (containing 100 ng of DNA) were taken from the reaction mixture at selected incubation times and inactivated by addition of 0.5% (w/v) SDS. The reaction products were separated by agarose gel electrophoresis in tris-acetate-EDTA buffer and stained using 0.5× of Sybr Gold. The agarose concentration used varied depending on the plasmid length (0.7% for the 7.2 kbp plasmid, 1.0% for 4.6 and 2.7 kbp, and 1.2% for 1.8 kbp). For the assays with fluorescently labelled intasomes, agarose gels were scanned with a Typhoon FLA9500 (GE Healthcare) at 532 nm. Intensity profiles of the gel bands were generated using scanning probe image processor (SPIP, v6.4, Image Metrology). Quantification of gel band intensities was performed using the peaks and baselines routine in Origin (OriginLab, Massachusetts, USA). Uncropped and unprocessed scans are supplied in the Source Data file.

**Kinetic modeling of ensemble integration assay**. The data of the strand transfer assays for the length dependence of integration (1.8, 2.7, 4.6, and 7.2 kbp plasmids) was globally fitted to the model depicted in Supplementary Fig. 4a. The fit used the means and SDs from three repeats for each plasmid length and minimized the error-weighted least-square deviation of the model from the data using the Levenberg-Marquardt algorithm implemented in the lsqnonlin function of Matlab (Mathworks). The fraction of active intasome, the rate of half-site integration ($k_{hs}$), and the rate of full-site integration ($k_{fs}$) were treated as fitting parameters that are the same for all plasmid length. In addition, the rates for TCC formation ($k_{on}$) were treated as four indepdent fitting parameters for each plasmid length. The off-rate for dissociation of the TCC was initially fit, but found to be very small or negligible. Therefore, it was fixed to $k_{off} = 10^{-10}$ min⁻¹ in the final analysis.

**Magnetic tweezers setup**. We used a custom-built MT setup wherein a pair of $5 \times 5 \times 5$ mm³ permanent magnets (W-05-N50-G, Supermagnete, Switzerland) oriented in vertical configuration[32] and with a gap size of 1 mm was employed to topologically constrain the DNA tethers. A DC-Motor (M-126.PD2, PI, Germany) controlled the distance between the flow cell and magnets, another DC-Motor (C-150.PD, PI, Germany) controlled rotation of the magnets. A ×40 oil-immersion objective (UPLFLN ×40, Olympus, Japan) was employed to image the beads onto a CMOS sensor camera (12 M Falcon2, Teledyne Dalsa, Canada) with a field of view of 400 µm by 300 µm. Images were recorded at 58 Hz and transferred to a frame grabber (PCIe 1433, NI, USA). A custom-written tracking software analysed the images to extract the $(x,y,z)$ coordinates of all beads in real time[33]. A LED (69647, Lumitronix LED Technik GmbH, Germany) was used for illumination. For tracking of the bead z-position, a look-up table (LUT) is required to translate the defocused pattern of the bead to its height. The LUT was generated over a range of 10 µm, with a step size of 100 nm, by moving the objective using a piezo stage (Pifoc P-726.1CD, PI, Germany).

**DNA constructs and magnetic beads for magnetic tweezers**. A 7.9 kbp DNA construct, prepared as described in published protocols [34,35], was used for all our MT measurements. PCR-generated DNA fragments (~600 bp) labeled with

multiple biotin and digoxigenin groups were ligated to the DNA, to bind magnetic beads and the flow cell surface, respectively. For all measurements, except for those that probe the force stability of the STC, we used 1.0 μm diameter MyOne magnetic beads (Life Technologies, USA). For force probing of the STC, we used 2.8 μm diameter M270 magnetic beads (Life Technologies, USA). The DNA construct was attached to the streptavidin-coated beads by incubating 0.5 μl of picomolar DNA stock solution and 2 μl MyOne beads in 250 μl ×1 phosphate-buffered saline (PBS) (Sigma-Aldrich, USA) for 5 min. Subsequently, the bead-coupled DNA constructs were introduced into the flow cell. Alternatively, 0.5 μl DNA stock solution and 10 μl M270 beads were incubated in 250 μl 1× PBS for 2 min, prior to their introduction in the flow cell.

**Magnetic tweezers flow cells.** Flow cells were built from two microscope coverslips (24 × 60 mm, Carl Roth, Germany). To attach the DNA molecules to the flow cell, the bottom coverslip was first coated with (3-Glycidoxypropyl)trimethoxysilane (abcr GmbH, Germany). Afterwards, 50 μl of a 5000× diluted stock solution of polystyrene beads (Polysciences, USA) in ethanol (Carl Roth, Germany) was deposited on the silanized slides, slowly dried in a closed container, and baked at 80 °C for 1 min, to serve as reference beads. The top coverslip was processed using a laser cutter, to produce openings with a radius of 1 mm, to enable liquid exchange. The two coverslips were glued together by a single layer of melted Parafilm (Carl Roth, Germany), precut to form a ~50 μL channel connecting the inlet and outlet opening of the flow cell. Following flow cell assembly, 100 μg/ml anti-digoxigenin (Roche, Switzerland) in 1× PBS was introduced and incubated for 2 h. To minimize nonspecific interactions, the flow cell was flushed with 800 μl of 25 mg/ml bovine serum albumin (Carl Roth, Germany), incubated for 1 h and rinsed with 1 ml of 1× PBS. The premixed DNA-bead solution was added to the flow cell for 5 min for MyOne beads or 2 min for M270 beads to allow for the digoxigenin–anti-digoxigenin bonds to the surface to form. Subsequently, the flow cell was rinsed with 2 ml of 1× PBS to flush out unbound beads. Next, the magnet was mounted, which constrains the supercoiling density of the tethers and applies an upward force on the beads. For measurements probing the force dependence of STCs, an additional step involving the application of high (50 pN) tension for 30 min was implemented to remove weakly bound tethers.

**Measurement protocols for magnetic tweezers experiments.** Prior to each measurement, selected beads were tested for the presence of multiple tethers and torsional constraint, by measuring their response to force and torque. The presence of multiple tethers was assessed by introduction of negative linking differences under high tension ($F \geq 5$ pN.) In the case of a single DNA tether, high tension impedes the formation of plectonemes at negative linking differences. As a result, no height change is observed. In contrast, for the case of multiple tethers, introduction of negative linking differences results in braiding, decreasing the $z$-extension of the bead. Beads bound by multiple tethers are discarded from further analysis. To assess whether DNA tethers were fully torsionally constrained, positive linking differences are introduced at low force (0.4 pN). In torsionally constrained DNA tethers, this leads to the formation of plectonemes, which decrease the $z$-extension. In nicked DNA tethers, no linking difference can be induced and the $z$-extension remains constant on magnet rotation.

Last, to define the state of zero torque ($Lk = Lk^0$) of individual DNA molecules relative to the position of the rotation motor, rotation–extension curves were constructed before the start of the single-molecule strand-transfer assay.

The single-molecule strand-transfer assay starts with the introduction of reaction buffer in the flow cell, followed by the application of a linking difference of ±25 in the tethers at 0.5 pN tension. The $z$-fluctuations of supercoiled DNA tethers in reaction buffer are recorded for ~1 h, to verify the absence of anomalous behavior. Next, the $z$-translation motors that control the magnet position were moved closer to the flow cell, to apply a force of 5 pN. CI (10 nM; 150 μL) is then flushed in the flow cell via the use of a peristaltic pump (flow rate ~150 μL min⁻¹). Consecutively, the force is reduced to its original value (0.5 pN). All experiments were performed in reaction buffer. For some experiments, Raltegravir (NIBSC labs) was added to the reaction buffer at a final concentration of 10 μM.

For force probing of STCs, the conventional single-molecule strand-transfer assay is executed and followed by a sudden force jump from 0.5 pN to high force (30 pN, 40 pN, and 50 pN). The response of the tethers was recorded for ~12 h. We only considered DNA tethers exhibiting a physical signature of intasome binding for our analysis. As a control, we tested the rupture probability distribution in the absence of intasome, under otherwise identical conditions, to evaluate the stability of the non-covalent interactions between the tether and the bead, respectively, of the flow cell surface. It was found that the mean rupture lifetime of the handles at 40 and 50 pN is more than one order-of-magnitude longer (8.2 ± 0.8 h at 40 pN; 2.5 ± 0.4 h at 50 pN), compared with the lifetime of the STC at that force. At 30 pN, the difference was only ~2-fold (22.7 ± 0.5 h) and the distributions of lifetimes overlapped more significantly. Therefore, our estimate of the zero-force lifetime of the STC might be slightly biased by this effect and constitutes a lower limit.

Torque probing of STCs starts with the conventional single-molecule strand-transfer assay using either positively or negatively supercoiled DNA tethers as a target. After ~2 h, the rotation magnet is driven to introduce a change in the linking difference. At intervals, the magnets are alternatingly rotated in opposite sense, to probe relaxation of the same molecules under a positive and negative torque.

**Analysis of magnetic tweezers data.** Real-time tracking was performed using the open source software framework based on the CUDA parallel computing framework developed previously[33]. This framework employs the Quadrant Interpolation algorithm to enable accurate and simultaneous tracking of many beads in parallel. Further processing of the MT data was carried out using custom-written Matlab routines. For unbiased step-finding, we used a published step-finder algorithm[24]. The extent of extension fluctuations $\sigma_z$ was calculated using a 0.5 s moving window. Lifetime and step-size distributions of experimental data are evaluated using a maximum-likelihood algorithm. Unless indicated explicitly, errors on fit parameters represent 95% confidence intervals.

**Estimation of energetic contributions in strand transfer.** We first estimate the free-energy release upon relaxation of supercoils. The free energy released by nicking of a supercoiled plasmid can be estimated using the empirical relationship previously determined:[36]

$$\Delta G_{SC} = \frac{1100}{N} \cdot k_B T \cdot \Delta Lk^2 \qquad (1)$$

where $N$ is the number of basepairs ($N = 4361$ bp in pBR322), $k_B$ the Boltzmann constant, and $T$ the absolute temperature ($T = 310$ K). For example, for pBR322 with supercoiling a density $\sigma = -0.05$ (corresponding to a linking difference $\Delta Lk = 20.75$), we estimate $\Delta G_{SC} \approx 110 k_B T$. For all conditions and plasmids used in bulk reactions in this work $\Delta G_{SC}$ is ≥40 $k_B T$.

Next, we estimate the free energy due to DNA bending in the active site. The free energy required for bending DNA can be estimated in a harmonic approximation[36] by

$$\Delta G = A \cdot k_B T \cdot \frac{(\theta)^2}{2l} \qquad (2)$$

where the $A$ is the bending persistence length ($A \approx 40$ nm under the conditions used), $\theta$ is the bend angle of the DNA, and $l$ is the length over which the DNA is bent. For the DNA in the TCC, we can estimate from the crystal structure[14] $\theta \approx 60°$ and $l = 14$ bp for the DNA strand engaged in the active site, which corresponds to $\Delta G \sim 4.6$ $k_B T$. In the context of our work, it is of interest to evaluate the effect of facilitated binding due to pre-bending in supercoiled DNA. To estimate the value of $\theta_{pre}$ in a supercoiled DNA, we use the simulation results by Ubbink and Odijk[37]. In supercoiled DNA with supercoiling density $\sigma = \pm 0.05$, the value of the pre-bending angle in a 14 bp dsDNA is $\theta_{pre} = 10°$, which corresponds to 0.13 $k_B T$ and reduces the energy penalty for target DNA bending to ~4.5 $k_B T$. Importantly, converting the fully double-stranded DNA in the TCC to nicked DNA in the STC (with minimal bending rigidity[38]) reduces the energy penalty for bending to close to zero, thus causing a reduction in free energy by ~4–5 $k_B T$.

Last, we quantify the free energy for binding at the apical interfaces under positive and negative torsional strain. The contribution of DNA binding at one of the apical interfaces in the STCs under the conditions of our measurement can be estimated from the times spend in the bound and unbound configurations as

$$\Delta G = k_B T \ln(\tau_{bound}/\tau_{unbound}) \qquad (3)$$

where $\tau_{bound}$ is the lifetime of the bound state and $\tau_{unbound}$ the lifetime of the dissociated state. Under negative torque, $\tau_{bound} \sim 110$ s, whereas for positive torque $\tau_{bound} \sim 67$ s. From the slope of the extension steps in the supercoil relaxation traces, we estimate the time to remove one turn ~4 ms and $\tau_{unbound} \sim 4$ ms $\langle \Delta Lk \rangle = 13.2$ ms (negative torque) or 15.6 ms (positive torque), where $\langle \Delta Lk \rangle$ is the mean step size (Fig. 4g). The estimates for $\tau_{bound}$ and $\tau_{unbound}$ correspond to $\Delta G = 9.0$ $k_B T$ for negative torque and 8.4 $k_B T$ for positive torque, respectively. We note that the difference in stabilization between positive and negative torque is only 0.6 $k_B T$ per interface or 1.2 $k_B T$ for the two apical interfaces. This relatively modest difference can be clearly detected in the sensitive single-molecule tweezers assay (Fig. 4g, h), but is too small to be reliably observed in our bulk assay with negatively and positively supercoiled DNA (Fig. 1i).

**Reporting summary.** Further information on research design is available in the Nature Research Reporting Summary linked to this article.

## Data availability
Data supporting the findings of this manuscript are available from the corresponding authors upon reasonable request. A reporting summary for this Article is available as a Supplementary Information file.

The source data underlying Figs. 1h, i, 2d, h, i, 3e, h, 4f, 5b, d and Supplementary Figs. 4d, 6a, b, 7d, e are provided as a Source Data file.

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

## Acknowledgements

We thank Peter Cherepanov for discussions and sharing expression plasmids of mutant PFV integrase; Gregory Van Duyne and Kushol Gupta for sharing coordinates of a PFV intasome SAXS model; Ellis Durner and Hermann Gaub for discussions; Thomas Nikolaus, Laura Krumm, and Flore Dewit for experimental help; Joachim Rädler and Tim Liedl for the use of AFM; and Steven De Feyter for continuous support. Funding: KU Leuven through IDO/12/08, the Deutsche Forschungsgemeinschaft through SFB 863 (Project A11), and The Funds for Scientific Research Flanders (FWO) for a post-doctoral grant to W.V.

## Author contributions

Conceptualization, W.V and J.L. Methodology, W.V. and J.L. Investigation, W.V., T.B., P.K. Resources, T.B., P.W., W.O., L.M., P.N., Z.D., J.L. Original draft, W.V., J.L. Funding acquisition, W.V., Z.D., J.L. Supervision, W.V. and J.L.

## Competing interests

The authors declare no competing interests.
