## [Peer Review File · Nature Communications]

Editorial Note: This manuscript has been previously reviewed at another journal that is not operating a transparent peer review scheme. This document only contains reviewer comments and rebuttal letters for versions considered at *Nature Communications*. Mentions of the other journal have been redacted.

Reviewers' Comments:

Reviewer #1:

Remarks to the Author:

The authors addresses my concerns.

Reviewer #2:

Remarks to the Author:

The paper is much improved by addition of experiments showing intasome activity via transfer of fluorescent viral mimic oligos.

I think the paper should be accepted with minor revisions, mainly clarifications of the following points, which the editor could deal with without requiring further reviewer approval.

1. Aspects of the model of the path of DNA along the intasome remain unclear. How is DNA in the "longitudinal" auxiliary binding surface connected to DNA bound to apical auxiliary surfaces, i.e. in Fig. 2 J and the rightmost schematic of Fig. S9, how are the 2 strands related to each other? In Fig S9, what is the significance of the small blue and red squares outside of the oval intasome? What is a "branch point"? If a "branch point" is where more than 2 DNA strands appear to cross (whether or not an intasome is there), was the possibility that intasomes stabilize multiple crossings taken into account in calculating the dependence of on-rate on the number of branch points (Fig S6F)? A little more explanation of how the kinetic model in Fig S6 was fit to data would be helpful.

2. Is "stabilization" of traces under reaction conditions assumed to be due to irreversible strand transfer events stabilizing DNA binding to auxiliary surfaces? If so, this could be stated more clearly. Would it then be expected that tethers "popped" open at high force would be uncoilable due to their containing a single stranded region? If "popped" tethers were coilable, would this imply strand transfer was reversible? Was this tested?

maybe DNA could be coilable after strand transfer of just one strand of viral end DNA mimic if in tethers "popping open" at high force the integrase remained associated with target DNA on one side of hemi-ligated viral mimic DNA but lost its "grip" on target DNA on the other side, allowing torsional relaxation and bead extension; but when force is reduced to test for coilability, the integrase "re-grabs" the released portion of target DNA, re-establishing torsional constraint. Then coilability would not imply that integration was reversible. But if adding raltegravir while the DNA was extended blocked subsequent coilability when force was reduced, this would argue that coilability after strand transfer (assuming it was observed) involved reversal of hemi-strand transfer, and hence that integration is reversible. Can the authors consider this scenario?

3. How was ΔW_r calculated in Fig. S16?

4. Clarify if "chemical deproteination" means treatment with SDS or something else, and if DNA was

purified after SDS treatment.

Reviewer #3:

Remarks to the Author:

The reviewer still has concerns that the new findings are not of sufficient significance to merit publication in Nature Communications. Admittedly, that is a value judgement. The bulk the literature indicates that STC intasomes are stable. Reference 16, reporting a lifetime of less than 1 second is an outlier. If intasomes were really that unstable the numerous published biochemical and structural studies would have been impossible.

Points-by-point response to reviewers' comments

The reviewers comments are repeated in blue italic and our comments are black.

Reviewer #1 (Remarks to the Author):

The authors addresses my concerns.

Reviewer #2 (Remarks to the Author):

The paper is much improved by addition of experiments showing intasome activity via transfer of fluorescent viral mimic oligos.

I think the paper should be accepted with minor revisions, mainly clarifications of the following points, which the editor could deal with without requiring further reviewer approval.

1. Aspects of the model of the path of DNA along the intasome remain unclear. How is DNA in the “longitudinal” auxiliary binding surface connected to DNA bound to apical auxiliary surfaces, i.e. in Fig. 2 J and the rightmost schematic of Fig. S9, how are the 2 strands related to each other? In Fig S9, what is the significance of the small blue and red squares outside of the oval intasome?

The symbols in Supporting Figure 2d correspond to the mean values of the DNA exit angles as determined experimentally from our high-resolution AFM images.

We have now added clarifications to the text in the corresponding figure legend, now Supplementary Figure 2D “Reconstruction of a mesoscale model for DNA folding in TCCs and STCs”. In addition, we have added lines to connect exit points of target DNA extruding as loops from the intasome (Figure 2J; Supporting Figure 5D)

What is a “branch point”?

Supercoiling branch points are defined as the points where the superhelix axes from more than two plectonemic segments intersect. We have added this definition in the figure legend of Supplementary Figure 4.

If a “branch point” is where more than 2 DNA strands appear to cross (whether or not an intasome is there), was the possibility that intasomes stabilize multiple crossings taken into account in calculating the dependence of on-rate on the number of branch points (Fig S6F)?

The relevant figure is now Supporting Figure 4. In panel F, we show the on-rate, which was determined from fits of the kinetic model to integration data with plasmids of different lengths, as a function of number of branchpoints. The number of branch points was obtained in completely separate experiments by performing extensive AFM imaging on (bare DNA) plasmids of different lengths. The numbers reported do not make any assumptions of how many branch points might be bound by the intasome.

However, the data suggest that capturing a single branch point is sufficient for formation of a functional target capture complex, as the on-rate levels off for > 1 branch point present.

A little more explanation of how the kinetic model in Fig S6 was fit to data would be helpful.

We have added a more detailed description of how the fitting to the kinetic model was performed in the methods section, specifically under “Kinetic modelling of ensemble integration assay”, we have added the following text:

“The global fit of the kinetic models was to the data for the 1.8, 2.7, 4.6, and 7.2 kbp plasmid strand transfer integration assays. The fit used the means and standard deviations from 3 repeats for each plasmid length and minimized the error-weighted least square deviation of the model from the data using the Levenberg-Marquardt algorithm implemented in the lsqnonlin function of Matlab (Mathworks). The fraction of active intasome, the rate of half-site integration (k_{hs}), and the rate of full-site integration (k_{fs}) were treated as fitting parameters that are the same for all plasmid length. In addition, the rates for target capture complex formation (k_{on}) were treated as four independent fitting parameters for each plasmid length. The off-rate for dissociation of the target capture complex was initially fit, but found to be very small or negligible. Therefore, it was fixed to $k_{off} = 10^{-10} \text{ min}^{-1}$ in the final analysis.”

2. Is “stabilization” of traces under reaction conditions assumed to be due to irreversible strand transfer events stabilizing DNA binding to auxiliary surfaces? If so, this could be stated more clearly.

We indeed provide strong evidence that the stabilization of the intasome is due to irreversible strand transfer events. Our interpretation is that upon strand transfer reaction, the complex is locked in place by the covalent attachment due to the irreversible strand transfer reaction, which means that the dynamic target search and sampling of different binding locations stops.

To clarify this point, we have added the following text:

“A second signature is sampling of different extension states followed by a stable extension level, consistent with covalent anchoring of the target DNA to the intasome on strand transfer that stops the dynamic target search.”

Would it then be expected that tethers “popped” open at high force would be uncoilable due to their containing a single stranded region? If “popped” tethers were coilable, would this imply strand transfer was reversible? Was this tested?

The referee raises an interesting question. Unfortunately, the coilability of tethers can not be tested at high forces, as DNA is not plectonemically coilable at forces > 1 pN due to melting of the DNA strands. We argue that integration is essentially irreversible since we never (0 out of ~ 200 traces) see tethers that go back to the dynamic fluctuations associated with dynamic target search after reaction. In addition, we do not observe a distribution of partially relaxed topoisomers in the bulk assays, but only intact supercoils and nicked or linearized plasmids, also strongly suggesting that there is no back reaction.

Maybe DNA could be coilable after strand transfer of just one strand of viral end DNA mimic if in tethers “popping open” at high force the integrase remained associated with target DNA on one side of hemi-ligated viral mimic DNA but lost its “grip” on target DNA on the other side, allowing torsional relaxation and bead extension; but when force is reduced to test for coilability, the integrase “re-grabs” the released portion of target DNA, re-

establishing torsional constraint. Then coilability would not imply that integration was reversible. But if addingaltegravir while the DNA was extended blocked subsequent coilability when force was reduced, this would argue that coilability after strand transfer (assuming it was observed) involved reversal of hemi-strand transfer, and hence that integration is reversible. Can the authors consider this scenario?

The referee is correct that there is unbinding and rebinding of the interfaces that ensure coilability – this is precisely what gives rise to the distribution of extension (and thus linking number) steps (see e.g. Fig. 4B), analyzed in Figure 4F.

We note that for coilability, it would make no difference whether one or both strands have reacted, since a single nick is sufficient for the relaxation of plectonemic supercoils, see e.g. the bulk assays reported in Supplementary Figures 1 and 4.

We argue that integration is essentially irreversible since we never (0 out of ~200 traces) see tethers that go back to the dynamic fluctuations associated with dynamic target search after reaction. In addition, we do not observe a distribution of partially relaxed topoisomers in the bulk assays, but only intact supercoils and nicked or linearized plasmids, also strongly suggesting that there is no back reaction.

3. How was ΔW_r calculated in Fig. S16?

The referee raises a good point. The analysis of linking number changes is in the plectonemic regime, such that the twist is constant and $\Delta W_r = \Delta Lk$. Nonetheless, to avoid confusion, we have relabeled the graphs on the figure (now Supplementary Figure 7) “ ΔLk ”.

4. Clarify if “chemical deproteination” means treatment with SDS or something else, and if DNA was purified after SDS treatment.

To purify the reaction mixture, we used a commercial PCR purification kit. We have added the following text to the methods section:

“For analysis of the DNA topology after reaction, we applied the additional steps of deproteinating the reaction mixture using a PCR purification kit (Zymo Research) followed by adjusting the salt concentration of the eluate to 125 mM NaOAc, 5 mM Mg(OAc)₂, 25 mM Tris-HCl (pH = 7.4), 10 μ M ZnCl₂.”

Reviewer #3 (Remarks to the Author):

The reviewer still has concerns that the new findings are not of sufficient significance to merit publication in Nature Communications. Admittedly, that is a value judgement. The bulk the literature indicates that STC intasomes are stable. Reference 16, reporting a lifetime of less than 1 second is an outlier. If intasomes were really that unstable the numerous published biochemical and structural studies would have been impossible.

We agree with the referee that a stable intasome is supported by several lines of evidence. However, our findings go much beyond just demonstrating a stable strand transfer complex. For example, we show that the stability of the final strand transfer complex –surprisingly!– is not the dominant factor driving the reaction forward – release of DNA supercoiling free energy is. In addition, our data identifies the relevant interfaces, provides evidence for an allosteric transition upon reaction, and demonstrate that multivalent target binding allows for irreversible yet dynamic target search.